# "When a story gets a face. . .": Visual elicitation of Serbian perspectives on Afghan refugee women's maternity experiences in Serbia

Esther SHARMA[1], Rebecca LABAN-SHARMAN[2], Baharak JAVAZIAN[2], Natasha HOWARD[1]*, Diane DUCLOS[1]

**1** Department of Global Health and Development, London School of Hygiene and Tropical Medicine, London, United Kingdom, **2** Independent Researcher, Belgrade, Serbia

* natasha.howard@lshtm.ac.uk

**Data Availability Statement:** The data set is available at LSHTM Data Compass DOI: 10.17037/DATA.00003389.

## Abstract

Serbia is a country through which many Afghans pass, en route to the European Union. Pregnancy and birth continues for Afghan women while journeying. This study aims to examine the understanding and attitudes of Serbian health professionals and members of women's civil society organisations (CS0) about the perinatal experiences of Afghan women in Serbia, using a webcomic to elicit responses. A total of 38 respondents completed the questionnaire, including health professionals (n = 10), women's CSO members (n = 6), and others (n = 10). The majority had little awareness of the experiences of Afghan women around the childbearing time and for most respondents, viewing the webcomic raised their awareness. Qualitative questionnaire data were analysed thematically, and four inductive themes developed: (1) maternal health provision as inadequate; (2) Afghan women face difficulties in Serbia; (3) solidarity with Afghan women; and (4) the webcomic raises awareness. Webcomics, as a visual modality, may play a valuable role in increasing empathy and awareness of refugee women's perinatal experiences among citizens.

## Introduction

In the 1990s, approximately 2.3 million refugees fled the former Yugoslav countries, seeking sanctuary in Western Europe, prompting suggestions that Europe was experiencing a refugee crisis [1]. Since 2015, the former Yugoslav countries (hereon in referred to as Western Balkan countries), have been on the receiving end during the recent so-called 'migrant crisis' (a term deserving of critique, see Almustafa [2]). Among all Western Balkans countries, Serbia hosted the largest number of migrants and refugees in 2022 [3]. However, refugees in Serbia do not generally view Serbia as a country in which to seek international protection on a long-term basis, but rather a country through which they transit to reach the European Union (EU) Schengen Zone. Afghans have consistently comprised one of the largest groups of migrant and refugee populations [3].

Reasons for Afghans' migration to Western Europe are longstanding and multifactorial. Afghans have a history of migration for trading, agriculture, and military reasons [4], resulting

**Funding:** This work was supported by the London School of Hygiene and Tropical Medicine (Public Engagement Small Grant Scheme to ES). The funders had no role in study design, data collection and analysis, decision to publish, or preparation of the manuscript.

**Competing interests:** The authors have declared that no competing interests exist.

in global networks, fostering and facilitating further migration for family and network (re)unification [5, 6]. Over the past fifty years, conflict in Afghanistan has exacerbated migration, with an estimated total of 2.4 million Afghan refugees registered with UNHCR at the end of 2021 [7]. For the purpose of this paper, we use the term 'refugee' to mean, ". . . people who have fled war, violence, conflict or persecution and have crossed an international border to find safety in another country" [8].

Legal routes to safety within the EU have been, and continue to be, very limited. For that reason, many Afghans have no choice but to seek international protection in the EU by travelling there irregularly by overland routes. These journeys, known colloquially as 'the game', are typically staged and protracted, using smugglers to assist with travelling [9]. The risks associated with 'the game' are high and gendered. Women who travel with smugglers are at risk of sexual exploitation, trafficking, and gender-based violence [10]. Western Balkan countries are a common route for Afghans travelling overland. The situation for refugees passing through Serbia has been variable since 2015, with previously *de facto* open borders to the EU now closed, resulting in refugees becoming 'stranded' in Serbia, and having to take greater risks to enter the EU. State-run accommodation (informally referred to as 'camps') is provided for those choosing to register with authorities on arrival in Serbia. Those choosing not to, live instead in informal squats or tents. Access to free emergency health care, including maternity care, is available to those in state accommodation, while healthcare for those living outside camps is restricted by the state to emergency treatment only.

Despite media portrayals of Afghan refugees entering the EU being young men, significant numbers of Afghan women have travelled overland through Western Balkan countries. Exact numbers are not known, due to UNHCR failing to collect adequate sex-disaggregated data. However, it is estimated that women and children make up 55% of all categories of migrants entering Europe, with 10% already pregnant [11]. Migration is transformative for women. A growing body of migration research recognises women as independent actors in migration and yet insufficient data results in their invisibility (and in relation to this study, erasure of maternal health needs) in policy responses [12–14].

The lived maternity experiences for refugees on the move, that is women who are transiting from one country to another, no matter how protracted that movement may be, is under-researched [15]. Most of this research has focussed on women in refugee camps [16–18].

This study is part of wider research exploring the perinatal experiences of Afghan women on the move in and through Serbia, and the experiences of various actors in Serbia supporting Afghan women during the perinatal period. This study aimed to understand the awareness and attitudes of Serbian health professionals and members of women's civil society organisations about the perinatal experiences of Afghan women in Serbia, using a webcomic to elicit responses.

## Methods

### Study design

We conducted a visual-elicitation study, using a co-produced webcomic as a medium through which to elicit responses to an online survey. Visual elicitation is a widely-used creative method for stimulating interview responses in qualitative research, with photographs the most common. Bagnoli argues that including the visual elements in research "which rely on other expressive possibilities, may allow us to access and represent different levels of experience." [19, p.547].

Our research question was: "What are the awareness and attitudes of Serbian health professionals and members of women's civil society organisations about the perinatal experiences of Afghan women in Serbia?"

## Webcomic

**Purpose of the elicitation tool.**    Webcomics use dual forms of text and visualisation and unlike written narratives, are textually light. This, combined with their formation as a series of frames, causes readers to become active agents, using their imagination to fill gaps and between text and frames, thus drawing readers to creating their own meanings [20]. The power of webcomics is their use of informal language, relatability, and accessibility [21], which are well-suited to communicating stories of human experience and phenomena [22]. They can aid communication of difficult topics—Chute notes that, "*Comics can express life stories, especially traumatic ones, powerfully because it makes literal the presence of the past by disrupting spatial and temporal conventions to overlay or palimpsest past and present*" [23, p.109].

**Theoretical stance.**    We took a decolonial feminist approach to visualisation of participants' lived experiences in the webcomic, seeking to centre participant voices, ". . . to recognize their [women of the "Global South"] sacrifices, honour their lives in all their complexity, the risks they took and the difficulties and frustrations they experienced. . ." [24, p.10]. Additionally, centring participants' voices in the webcomic challenges the visual tropes of non-white refugee women: the suffering body, passive victim, primitive 'other,' and breeding machine, as can be instrumentalised by the media and humanitarian organisations [25–27]. To this end, we planned to co-produce the graphic with interview participants, employing participatory research approaches for those with lived experiences, to shape it according to what *they* perceived as important in this visual re-telling of shared experiences, and to ensure it accurately reflected *their* collective experiences [28]. In addition to participation, co-production embodies collaboration, power sharing and equality [29]. However, conducting remote co-production during the Covid-19 pandemic with a group of highly mobile participants living in precarity, and already disproportionately affected by Covid-19, was challenging and we had lower engagement than anticipated. Participating in co-production can be burdensome, even more so when it involves recollecting painful memories [28], so we were cautious not to unduly pressure interview participants to join this additional research.

**Co-production and testing.**    We developed the webcomic over a one-year period. Data generated during 'field' visits to Serbia and interviews informed webcomic development. ES visited the spaces occupied by Afghan women, had informal conversations with Afghan women, in addition to non-governmental organisation (NGO) and state actors, and health professionals, working with Afghan women. She conducted narrative interviews with 11 Afghan women, face-to-face or remotely, to explore their experiences of pregnancy or birth in Serbia, with the assistance of a translator. Narrative interviews were employed as they enable participants to actively reconstruct their experiences, providing an insight into meaning-making in the context of displacement [27, 30]. A topic guide was developed as an aide memoire, starting with a broad opening question, "Can you tell me what it was like to be pregnant or give birth in Serbia?," but conversation was guided by participants as far as possible. Participation was anonymous and confidential. All participants gave informed consent after a full discussion about involvement and received a small token of appreciation.

Afghan women who had been interviewed were invited to join co-production of the webcomic. Five of 11 interview participants consented. We developed a composite narrative, based on preliminary analysis of narrative interviews, and common threads running through interview participants' collective experiences. Following script development, we engaged an artist

to create images for the webcomic based on the script and a set of reference photographs. The artist developed an initial set of images for the central character, "Zohra" and then a draft version of the webcomic. Care was taken throughout creation of images to ensure they would have meaning across cultures. The draft webcomic was translated into Dari (the most widely-spoken language in Afghanistan) and sent to the five women participating in the co-production, with a set of specific questions about their overall impression of the webcomic, it's storyline and Zohra's appearance.

Participants were asked to comment, either by text or voice message, including suggestions about any aspect of the webcomic and offered an opportunity for additional debrief afterwards if they wished. Some women included emotional responses to viewing the draft webcomic; "*Very wonderful and painful*" and "*Zahra has many problems in this story, which I have experienced them all!*". Their feedback confirmed that the artist had appropriately depicted Zohra's appearance. For example, one woman commented on Zohra's traditional clothing, "*In my opinion, the dress that Zohra is wearing is appropriate in the [refugee] camp because I used to wear the same dress in many camps in Serbia, there is no security for women and children*". On overall layout, women indicated there was a lack of flow between images and the text, which prompted a rearrangement to ensure that the webcomic flowed for those reading it either from right to left or left to right in any of the three languages in which it was produced. Additional practical layout considerations improved ease of readability on mobile devices.

Once alterations were made, the finalised English and Dari drafts were sent to participants for checking. The final English version was then translated into Serbian (Fig 1).

## Online survey

**Tool development.** An online descriptive exploratory survey was chosen as data collection method, including both closed- and open-ended questions, developed from existing literature on refugee women's perinatal experiences [15], knowledge of the Serbian maternity and

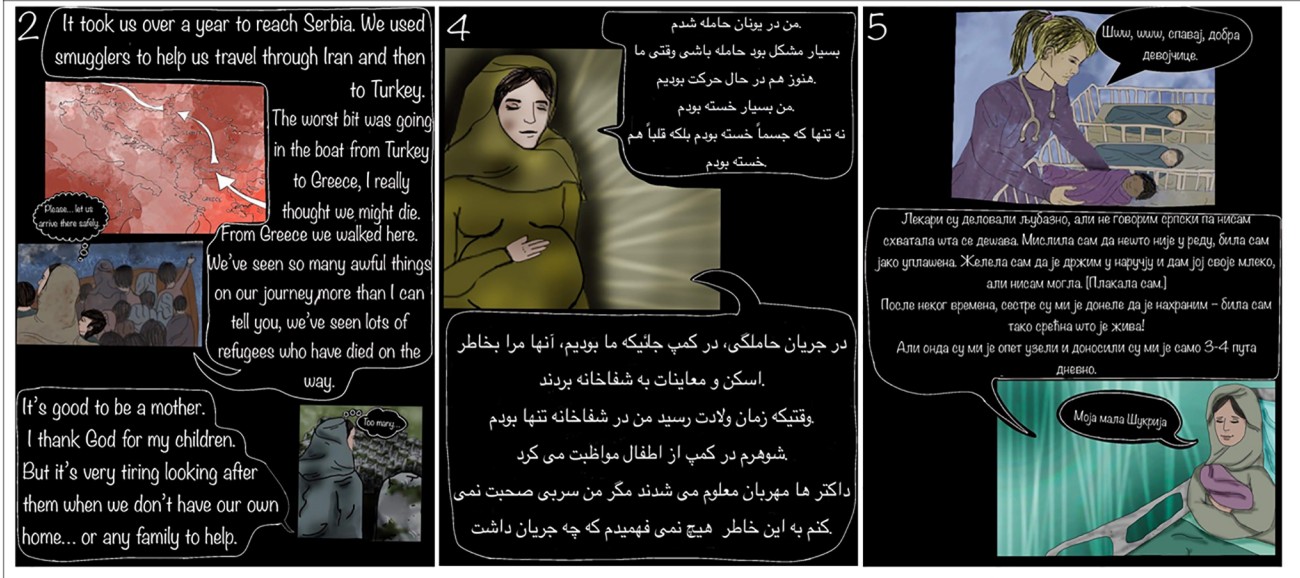

**Fig 1. Example webcomic frames in Dari, English and Serbian.** *NB*: The webcomic can be viewed in all three languages at https://birthingontheway. wixsite.com/project/webcomic-english.

health care system and preliminary analysis of field observations, conducted as part of the wider study. Firstly, this method offered the advantages of being able to capture the views and opinions of a large sample in a shortened time period. Secondly, it provided fewer barriers to participation by taking less time to participate than an interview, plus guaranteeing anonymity. Finally, it enabled us to contact health professionals remotely by email or phone messaging, in a context in which it was difficult to gain physical access to them in health facilities or recruit their participation in research.

The survey was developed in English and translated professionally into Serbian. Survey questions used Likert-scales and free text. Respondents were first asked to rank seven aspects of maternity care (i.e. provision of interpreters, respectful maternity care, information about how to access maternity care, regular antenatal checks, choices about what happens during labour and birth, support with breastfeeding, screening for mental health problems), from 1 (not at all important) to 10 (extremely important). The remaining free-text questions focused on: (1) participant awareness of Afghan women's experiences in Serbia around the childbearing time, pre- and post-webcomic viewing; (2) surprising and unsurprising aspects of Afghan women's experiences as depicted on the webcomic; and (3) opinions on the importance of Serbian maternal health providers understanding of these experiences.

**Participant sampling and recruitment.** We used a mix of purposive and convenience sampling, given the sampling constraints of online surveys [31]. We invited two groups of potential participants in Serbia to complete the survey, between 12 September and 31 October 2022: Serbian health professionals and Serbian women's civil society organisations (CSOs). The rationale for selecting these two groups was due to their common interest in issues pertaining to maternity care in Serbia, whether from a professional or service user perspective. RL-S contacted Serbian health professionals directly to describe the study and provide the link to webcomic and survey, while ES emailed Serbian women's CSOs requesting they distribute the link among their networks.

**Data collection.** We collected data by hosting the survey on Online Surveys [32], a secure web platform. We included a participant information sheet and consent form on the questionnaire landing page, requiring all participants to give informed consent before participation. Participants were then asked to view the webcomic prior to completing the 30-minute survey. Open-ended responses were professionally translated into English for analysis. None of the authors had access to information that could identify individual participants during or after data collection.

**Analysis.** ES analysed data from closed-ended questions descriptively, estimating frequencies and cross-tabulations using Excel. ES imported translated qualitative responses to NVivo v.12 [33] and analysed them as a whole, rather than individually, to develop themes holistically. We used reflexive thematic analysis, as described by Braun & Clarke [34]. This involved re-reading of responses for data familiarisation, initial data coding, searching for and reviewing themes from the codes, and finally refining and naming them. Both responses to questions and the ways the webcomic elicited responses were analysed.

## Ethics

Research ethics committees at the London School of Hygiene and Tropical Medicine (reference 22641–1) and University of Belgrade in Serbia (reference 1322/II-7) provided ethical approval. Informed consent was obtained from all participants using an online consent form which was embedded into the online questionnaire.

## Results

### Participant characteristics

Table 1 categorises the 38 individuals who viewed the webcomic and completed the online survey. Most were health professionals [22]. Of 10 identifying as 'other', one stated they were a woman's worker, another a mother, and the remainder did not indicate a professional identity. Six participants indicated they were from a woman's CSO.

### Descriptive statistics

Most (27; 71%) described themselves as having no or little awareness of the experiences of Afghan women around the childbearing time, prior to viewing the webcomic, and the majority (31; 82%) said their awareness had increased quite a lot or a lot. Just over half of the respondents (20; 53%) thought that it is very important that maternal health providers understand the experiences of Afghan women around the childbearing time, with all others except one stating that it is quite important for Serbian maternal health providers to understand these experiences (17; 45%).

All except two respondents ranked all seven aspects of maternity care at 6 of 10, or above, suggesting these participants considered all aspects listed in the questionnaire as important. Respectful maternity care ranked highest overall, with a mean of 8.2 out of 10, with screening for mental health problems and support with breastfeeding were both ranked lowest, with a mean of 6.3. The results are summarised in Table 2.

### Analytical themes

Thematic analysis identified four inductive themes: (1) maternal healthcare provision as inadequate; (2) Afghan women face difficulties in Serbia; (3) solidarity with Afghan women; and (4) the webcomic raises awareness.

**Maternal healthcare provision as inadequate.** Overall, respondents across categories described maternal health provision in Serbia for Afghan women as inadequate. The health system itself, language barriers, treatment of women by medical staff and poor postnatal care

**Table 1. Categorisation of survey participants.**

| Respondent type | Total |
| --- | --- |
| Health professional | 22 (58%) |
| Other | 10 (26%) |
| Woman's CSO member | 6 (16%) |

**Table 2. Rankings of importance of aspects of maternity care.**

| Aspect of maternity care | Mean rank (out of 10) |
| --- | --- |
| Respectful maternity care | 8.2 |
| Regular antenatal care | 7.4 |
| Provision of interpreters | 7.4 |
| Information about how to access maternity care | 7.1 |
| Choices about what happens during labour and birth | 6.8 |
| Screening for mental health problems | 6.3 |
| Support with breastfeeding | 6.3 |

were commonly cited examples. Several respondents indicated that the Serbian health system was not organised to adequately accommodate the needs of refugee women.

> *"Our healthcare system is not adapted to any foreign women, not just women in this situation who are particularly and additionally vulnerable."*

(Serbian CSO member)

There was also surprise expressed that women were able to access the healthcare system at all. For example, in response to a question, "When you read the webcomic, what surprised you most about the experiences of Afghan women in Serbia during the childbearing time?", one respondent answered, "*That they even have medical care.*" (respondent identifying as 'other').

Language barriers were perceived by many as particularly challenge for Afghan women. Being able to communicate with health professionals was considered by respondents as enabling women to understand what was happening to them, as well as a "*surmountable*" challenge, through the routine use of interpreters.

> *"I suppose the language barrier is the biggest challenge. It would be a lot easier for them if they could communicate with the healthcare workers who are helping them."*

(health professional)

In the webcomic, the main character, Zohra, describes how she was alone when she gave birth, and her worries and sadness that her baby was removed from her after birth, being brought to her three to four times throughout the day for feeding. An image is shown of a health professional placing the baby in a cot in a nursery, away from Zohra. Several participants commented on the attitudes of health professionals to women and disrespect during childbirth, not only for Afghan women, but for Serbian women as well. A woman's worker commented on the webcomic saying, "*The doctors seemed nice! Apart from a few exceptions, they are never nice.*" (woman's worker). This was illustrated further by another respondent:

> *"Serbian doctors often don't communicate with Serbian pregnant women, new mothers and women who have undergone gynaecological surgery of some sort either. Many of us have some kind of trauma because of that (among other things). We expect information but we don't get it. I cannot imagine what it's like for women who can't even expect information and what trauma they have because of that."*

(Serbian CSO member)

Responses such as these were not limited to those outside the health system. Health professionals themselves commented on the attitudes of their colleagues, for example:

> *"I suppose I should've been surprised by the medical staff's behaviour, unfortunately I wasn't."*

(health professional)

However, some health professionals acknowledged the inadequacy of the situation and expressed hope that it would change, and a need for greater awareness of the topic. For example:

"*I'm very sorry about the current situation and I hope it will change soon. Every woman deserves to be treated humanely, both during pregnancy and after childbirth. Anything other than that is a defeat of the entire society and a disgrace for all of us. Thank you for educating me on this subject, I wasn't even aware of how unfamiliar I was with what these women go through.*"

(health professional)

In terms of maternal health care itself, several respondents said that they thought the care for Afghan women was no different to that of Serbian women, specifically citing a lack of decision-making in childbirth, lack of postnatal care for women, including little breastfeeding support, and insufficient attention paid to maternal mental health. This is exemplified in the following quote:

"*A big problem is the fact that new mothers have no choice or options during and after labour because others decide for them (in what position they will give birth, whether they will use medicines or some other pain relieving methods, whether they will be with their child and breastfeed it or if they will give it formula, how and when the umbilical cord will be cut, whether the baby will be given a bath immediately or if it will have skin-to-skin contact and first breastfeeding. . .*"

(respondent identifying as 'other').

**Afghan women face difficulties in Serbia.** Respondents frequently commented on the difficulties for Afghan women in Serbia, notably perceptions of poor living conditions, material hardship, and challenges being in a country away from home. Respondents commented on poor living conditions as being inadequate for pregnant women, or mothers with children, also citing unsanitary conditions and risks of abuse of women.

"*The conditions in camps are equally bad for a new mother and for raising children.*"

(health professional)

Connected with poor living conditions was Afghan women's material hardship and poverty, the latter of which one respondent said "*. . . saddened me the most.*" *(health professional).* Other respondents highlighted the lack of basic necessities available to Afghan women, such as nappies, clothes, and food, or the money to purchase these items, with suggestions that items like these should be provided for mothers and their infants in the form of "*packages*" or "*funds*".

Being away from home and family was perceived by many as a source of stress for Afghan women in Serbia, with some commenting on the challenges of dealing with pregnancy and motherhood alone.

"*I believe that pregnancy and birth are much more stressful for them than for women who give birth in their own homeland.*"

(respondent identifying as 'other')

One respondent compared his experience of becoming a parent in Serbia with women giving birth away from home:

*"The fact that I'm a father of four means that I know what childbirth means even in ideal conditions, on the one hand, and on the other hand facing the possibility of a pregnant woman finding herself in a foreign environment, without a home, without help, without knowing the language. . . "*

(health professional)

These difficulties and hardships were largely viewed as situations which Afghan women should not have to face and were concerning for many respondents, whilst some also recognised women's agency:

"*Kudos to them for daring to have children as refugees, without a home and people they know*!"

(women's worker)

**Solidarity with Afghan women.**    The theme of solidarity with Afghan was expressed through respondents' personal experiences of interacting with Afghan women during the childbearing time, or through their desire to be able to provide assistance, "*. . . because they are people, like us*" (respondent identifying as 'other'). Another respondent expressed solidarity as:

"*. . .a sense of belonging regardless of different cultures, religions etc. should be nurtured.*"

(health professional)

In a vivid account of sharing a room with an Afghan woman after giving birth, a respondent described how she helped the woman during her hospital stay:

"*. . . I gave her painkiller suppositories, diapers, everything I had. She hugged me when I was leaving, I have a different view of the situation after that experience. I put myself in her shoes, I don't know whether I'd be able to handle it. I remember, when I gave her the suppositories, the nurses pushed her onto the bed and applied the suppository in front of the whole room because they couldn't explain to her how it was applied.*"

(Serbian CSO member)

Shared humanity, a desire for Afghan women to receive their rights in Serbia and expressions of wanting to help were common expressions of solidarity.

**The webcomic raises awareness.**    As mentioned, most respondents had little or no prior awareness of the experiences of Afghan women in Serbia around the childbearing time. However, several commented on how the webcomic increased their awareness of this matter, and its specific value as a mode of communication that enabled them to gain understanding of the key issues:

"*The comic is attention-grabbing and puts us in a real situation where we empathise with the actors, in this case Afghan women in a difficult position in life. The comic has largely helped to raise my awareness.*"

(health professional)

Another respondent commented on a particular aspect of the webcomic which made it relatable:

*"There already was some awareness of the problems of refugees but when a story gets a face, a first and last name, then it's no longer a problem someone somewhere is facing, rather it becomes a tangible thing and makes you think."*

(health professional)

For some health professionals, this increased awareness enabled them to have a better understanding of the issues facing Afghan women around the childbearing period, linking this increased understanding with being able to provide improved care. For example:

*"To help a patient, you must understand and empathise with them. How can you provide empathy and understanding, support, if you don't even know what the patient is going through?"*

(health professional).

## Discussion

These findings highlight firstly, the value of raising awareness about Afghan refugee women's perinatal experiences and that secondly, a webcomic can be an effective medium to elicit responses to a questionnaire to understand respondents' perceptions, attitudes and opinions. Among 38 respondents, existing awareness of the situation for Afghan women in Serbia during the perinatal period was explored, along with the extent to which the webcomic altered this awareness, aspects of the webcomic which were surprising and unsurprising, perceived challenges for Afghan women in Serbia, and the importance of various aspects of maternal health care. Overall, the webcomic can be seen to raise awareness of the childbirth and maternity experiences of Afghan women 'on the move' among its viewers, who also reported their previous awareness to be limited. Respondents particularly commented on the inadequacy of the maternal health care for Afghan women (but also Serbian women), the difficulties faced by Afghan women in Serbia, the lack of material provision for them, but also expressed solidarity with them.

The findings of this study, with respect to quality of maternal health care in Serbia, corresponds with other studies which have explored Serbian women's experiences of utilising maternity care. Lack of communication and informed consent during childbirth, for Serbian women using public maternity services, were identified in several studies, as was being without a companion during birth [35, 36] This points to a service in which respectful maternity care is not prioritised. Respectful maternity care is an essential element of quality maternity care and includes the tenets of the individual's right to dignity and respect, information and choice, in addition to evidence-based clinical care [37]. To date, there is a dearth of literature pertaining specifically to maternity experiences for refugee women in Serbia. However, the health service overall is an under-funded post-socialist health service which was not set up to respond to the needs of migrants and refugees [38, 39]. Some maternity service users engage with private care providers, enabling them to have a known caregiver and preferential treatment [40], but this requires sufficient financial means to pay for such a service, restricting this as an option for many refugees.

Breastfeeding is more commonplace among women in Afghanistan than women in Serbia. Data show that women 58% of women in Afghanistan (in 2015) were exclusively breastfeeding their infants by six months of age, compared with 13% of women in Serbia (in 2014) [41]. The World Health Organization guidelines on promoting breastfeeding within maternity facilities recommends rooming-in and demand feeding, in order to aid the establishment of breastfeeding [42]. This is further highlighted in the guidance for infant feeding in emergencies [43] in recognition of the fact that breastfeeding is particularly important in situations where there is lack of access to safe conditions for the preparation of formula milk [44], as can be the case in refugee camps. However, practices on postnatal wards in Serbia are not universally supportive of breastfeeding [45, 46]. For example, separating mother and baby during the initial postnatal period is at odds with rooming-in (keeping the mother and baby together to promote breastfeeding), and the common enforcement by health professionals of timed or restrictive breastfeeding, which can hinder a mother's milk supply, are commonplace on postnatal wards [46].

There is an urgent need for further research exploring the maternity experiences for all women in Serbia, as well as research focussed on the specific needs of refugee women, in order for systems fostering respectful, women-centred high-quality care to be developed. In Serbia, a medical-led, technocratic model of birth is followed [40]. That is, a philosophy of care viewing the birth process as a medicalised production line rather than a holistic process in which women are decision makers in their care [47]. Midwifery care is widely recognised as an essential component of the provision of quality maternal and newborn health care [48, 49]. A state-level investment midwifery education and incorporation of midwives into the Serbian health system could therefore play a pivotal role in improving quality of care for women using Serbian maternity services.

Solidarity towards Afghan women on from Serbian women on postnatal wards through the sharing of clothes and nappies was depicted in the webcomic. At a basic level, solidarity can be thought of as "... *an agreement of feeling or action*" [50]. Solidarity with Afghan women was invoked by questionnaire respondents in pragmatic terms—actions which can be taken to improve the situation for Afghan women in Serbia around the time of childbirth, and social terms and the shared humanity between refugees and citizens [51]. This solidarity is important to recognise in a context when there is an increasing anti-migrant sentiment in Serbia, contrasting with the Serbian state's desire to portray Serbia as welcoming to refugees, during the so-called 'migrant crisis' of 2015–16 [52]. One of the recommendations of a report which highlights the growth of far-right anti-immigrant sentiment and groups in Serbia, is the media's role in accurate reporting of issues facing refugees and migrants, including talking to the migrants and refugees themselves, as a means of countering disinformation and fear mongering among Serbian citizens [52]. With this in mind, webcomics may have a role in countering anti-migrant narratives.

As discussed earlier, webcomics, combining the visual with the textual, are well-suited to communicate challenging and traumatic subject matters [21]. They can "... offer a window into the subjective realities..." of others' lived experiences, providing an alternative means of relatedness, exploration and connection on an emotional level, with complex topics [53, p.5], and are a vehicle to creating an emotional connection between the narrative and viewer. It cannot be assumed that one short webcomic can fully encapsulate the wide-ranging experiences of Afghan refugee women who are pregnant or give birth while on the way through Serbia. Nonetheless, this study suggests that it is an advantageous approach to disseminating and giving visibility to Afghan women's experiences in a format that engages a wider audience–in this case Serbian health professionals and members of women's CSO's. In this study, many Serbian health professionals recognised a greater need for awareness of the issues facing Afghan refugee women in the perinatal period and noted that the webcomic provided them with a greater

understanding of these issues. Webcomics can be valuable in the education of healthcare professionals, acting as a powerful modality for conveying the lived experience of others [21, 53]. Further research could explore the use of webcomics as a tool for training health professionals in providing care to refugees.

## Limitations

This study was conducted in a relatively short period, thus limiting the number of responses obtained. The reasons for this were practical and budgetary, but a larger number of respondents would have provided more data and potentially more perspectives for consideration. Due to the questionnaire being self-administered, it is possible that respondents were those interested enough to complete the questionnaire having viewed the webcomic, thus introducing sampling bias. Finally, only 6 of respondents self-identified as being a member of a woman's organisation versus 22 being health professionals. Engaging with more women's organisations may have increased the proportion of respondents from this group.

## Conclusion

Among the respondents of this study, there was little previous awareness of the experiences of Afghan women travelling through Serbia around the childbearing time. The webcomic was a valuable modality in bringing Afghan women's experiences perinatal experiences to light. Whilst webcomics do not address the structural realities that cause refugee women to find themselves in these situations in the first instance, they may play a role in fostering solidary among Serbians and increasing awareness and empathetic care among health professionals.

## Acknowledgments

We would like to express our deep appreciation for the Afghan women who gave their time and shared their experiences. We are also very thankful to the Afghan women who took the time to read and provide feedback on the webcomic, as well as everyone who completed the survey. Finally, we are indebted to Shae White at the Real Birth Company for creating the webcomic illustrations.

## Author Contributions

**Conceptualization:** Esther SHARMA.

**Formal analysis:** Esther SHARMA.

**Funding acquisition:** Esther SHARMA.

**Investigation:** Esther SHARMA, Rebecca LABAN-SHARMAN, Baharak JAVAZIAN.

**Methodology:** Esther SHARMA.

**Supervision:** Natasha HOWARD, Diane DUCLOS.

**Writing – original draft:** Esther SHARMA.

**Writing – review & editing:** Natasha HOWARD, Diane DUCLOS.

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
