## [Decision Letter · Decision Letter 0]

12 Sep 2023

PGPH-D-23-00818

“When a story gets a face…”:  visual elicitation of Serbian perspectives on Afghan refugee women’s maternity experiences in Serbia.

Dear Dr. Howard 

Thank you for submitting your manuscript to PLOS Global Public Health. After careful consideration, we feel that it has merit but does not fully meet PLOS Global Public Health’s publication criteria as it currently stands. Therefore, we invite you to submit a revised version of the manuscript that addresses the points raised during the review process.

We look forward to receiving your revised manuscript.

Kind regards,

Wanga Zembe-Mkabile, Ph.D.

Academic Editor

Journal Requirements:

2. Some material included in your submission may be copyrighted. According to PLOS’s copyright policy, authors who use figures or other material (e.g., graphics, clipart, maps) from another author or copyright holder must demonstrate or obtain permission to publish this material under the Creative Commons Attribution 4.0 International (CC BY 4.0) License used by PLOS journals. Please closely review the details of PLOS’s copyright requirements here: PLOS Licenses and Copyright. If you need to request permissions from a copyright holder, you may use PLOS's Copyright Content Permission form.

Potential Copyright Issues:

Fig 1.tif: Please confirm whether you drew the images / clip-art within the figure panels by hand. If you did not draw the images, please provide (a) a link to the source of the images or icons and their license / terms of use; or (b) written permission from the copyright holder to publish the images or icons under our CC-BY 4.0 license. Alternatively, you may replace the images with open source alternatives. See these open source resources you may use to replace images / clip-art:

- https://openclipart.org/

Reviewer 1:

Review Comments to the Author

Thank you for the opportunity to review this article on a very important, global and topical issue. Please find a few comments below for your consideration. Could the authors please address the following;

1. I was intrigued by the use of a likert scale with such a wide range of responses (10 in total). As you know, the use of likert scales has its challenges in terms of weighting/ and or the meaning of responses. It is unclear in the article how such an analyses was handled to arrive at the scoring and ranking of responses presented in Table 2. It seems mean scores were calculated but an explanation is not given on why means were the most appropriate measure to report. I am sure the authors know that a choice has to be made between reporting modes, medians and frequencies vs. means and standard deviations. The appropriate analyses is informed by the design considerations re: how the likert scale was structured.

2. In the analysis, authors mention “cross tabulations”. It is unclear what cross tabulations were done, what test was this and the results thereof? It is unclear from the results of the quantitative data presented. Could authors classify- cross tabulations of what/which variables? See line 239-240

3. In line 273-272, the finding around mental health is an important aspect that needs to be highlighted in the discussion. The perinatal period is such a period in which many women develop depression even without the extended vulnerabilities of being a refugee, thus mental health of these women would have been hugely affected. Worth discussing how it is one of 2 categories that received the lowest ranking- perhaps something not very well understood by this population? Or it may speak to possible resilience and or coping mechanisms that, the lives lived by these women presents a constant and everyday worry that everything has become so internalised that they are not able to “amplify” the effects of mental health in a questionnaire? This goes back to my earlier comment about the likert scale and its ability to pick/” weight” particular issues.

4. Re: qualitative findings, I had difficulties following some quotes and drawing meaning out of them as the authors sometimes put down a section of the quote and not a full one. for example, in line 359, it will be good to see more of the text/quote leading to "...saddened me the most"

5. I found a few typos in the manuscript – e.g., see lines 455, 480, 526

6. Re: limitations section, I find the fact that the survey was online (line 231-232) another form of selection bias- only those who had online access were able to consider/complete the survey. Authors should include this in their limitations section.

6. PLOS authors have the option to publish the peer review history of their article (what does this mean?). If published, this will include your full peer review and any attached files.

 

Reviewer 2

Comments to the Author

This is a very important study on the maternity experiences of a vulnerable population of Afghan women. The manuscript is well written and formatted. Although the study was generally well conducted, given that the researchers were working with vulnerable populations, there are a few important details that are missing from the manuscript. My specific comments are as follows:

Methods:

• You used a decolonial feminist approach to visualize participant lived experiences. Given that you interviewed Afghan women, does approach align with their mindset? Was this approach also used to analyse the data from the health professionals and community members?

• Lines 147-148: Please indicate the exact period when the webcomic was developed and tested

• Lines 146-159: Please indicate the criteria that were used to select the 11 Afghan women that were interviewed e.g Afghan women who had recently delivered in Serbia. Please also briefly profile the women that were involved in the production of the comic. It is also not clear if the 5 participants that co-developed the comic were similar or different from the 6 other participants that only participated in the interview

• The study interviewed Serbian CSOs. Do Afghan civil society groups (formal or informal) exist in Serbia? If yes, then it would have also been good to include them in the study.

• Lines 219-228: In the introduction you indicated that registered Afghan refugees can access a range of healthcare services including maternity care. Unregistered refugees living in squatter camps can only access emergency care. Given the different types of healthcare services registered vs unregistered refugees can access, it would be good for you to describe the types of healthcare professionals that were interviewed in this study. Did these healthcare professionals work in maternity wards? Adding this information will contextualize the results

• Line 242: In the methods section you indicated that you used reflexive thematic analysis and that ES developed the themes. Did the other authors also analyse the data and interpret the findings? Were the same researchers involved in the development of the web-comic and analysis of the data?

Discussion

• Line 425-428: This study aimed to understand the awareness and attitudes of Serbian health professionals and members of women’s civil society organisations about the perinatal experiences of Afghan women in Serbia, using a webcomic to elicit responses. Therefore I suggest that you move line 434 to the first few sentences of the discussion (lines 425-428) as it highlights that most of the respondents were not aware of Afghan women's maternity experiences

• lines 513-521: There is potential selection bias because respondents needed access to an online survey. This study only asked questions on the healthcare experiences of Serbian women. Based on the results, it seems some of poor healthcare practices are experienced by all women in Serbia. Therefore the study could have benefited from asking general questions on maternity care in Serbia to contextualize the results

• Limited information is provided on the study participant characteristics. Additional information on who was included in the study (e.g. type of healthcare professional, maternal age, pregnant or postpartum woman etc.). would contextualize the results

Reviewers' comments:

Reviewer's Responses to Questions

**Comments to the Author**

1. Does this manuscript meet PLOS Global Public Health’s publication criteria? Is the manuscript technically sound, and do the data support the conclusions? The manuscript must describe methodologically and ethically rigorous research with conclusions that are appropriately drawn based on the data presented.

Reviewer #1: Yes

Reviewer #2: Yes

2. Has the statistical analysis been performed appropriately and rigorously?

Reviewer #1: I don't know

Reviewer #2: N/A

3. Have the authors made all data underlying the findings in their manuscript fully available (please refer to the Data Availability Statement at the start of the manuscript PDF file)?

Reviewer #1: Yes

Reviewer #2: Yes

4. Is the manuscript presented in an intelligible fashion and written in standard English?

Reviewer #1: Yes

Reviewer #2: Yes

5. Review Comments to the Author

Reviewer #1: Thank you for the opportunity to review this article on a very important, global and topical issue. Please find a few comments below for your consideration. Could the authors please address the following;

1. I was intrigued by the use of a likert scale with such a wide range of responses (10 in total). As you know, the use of likert scales has its challenges in terms of weighting/ and or the meaning of responses. It is unclear in the article how such an analyses was handled to arrive at the scoring and ranking of responses presented in Table 2. It seems mean scores were calculated but an explanation is not given on why means were the most appropriate measure to report. I am sure the authors know that a choice has to be made between reporting modes, medians and frequencies vs. means and standard deviations. The appropriate analyses is informed by the design considerations re: how the likert scale was structured.

2. In the analysis, authors mention “cross tabulations”. It is unclear what cross tabulations were done, what test was this and the results thereof? It is unclear from the results of the quantitative data presented. Could authors classify- cross tabulations of what/which variables? See line 239-240

3. In line 273-272, the finding around mental health is an important aspect that needs to be highlighted in the discussion. The perinatal period is such a period in which many women develop depression even without the extended vulnerabilities of being a refugee, thus mental health of these women would have been hugely affected. Worth discussing how it is one of 2 categories that received the lowest ranking- perhaps something not very well understood by this population? Or it may speak to possible resilience and or coping mechanisms that, the lives lived by these women presents a constant and everyday worry that everything has become so internalised that they are not able to “amplify” the effects of mental health in a questionnaire? This goes back to my earlier comment about the likert scale and its ability to pick/” weight” particular issues.

4. Re: qualitative findings, I had difficulties following some quotes and drawing meaning out of them as the authors sometimes put down a section of the quote and not a full one. for example, in line 359, it will be good to see more of the text/quote leading to "...saddened me the most"

5. I found a few typos in the manuscript – e.g., see lines 455, 480, 526

6. Re: limitations section, I find the fact that the survey was online (line 231-232) another form of selection bias- only those who had online access were able to consider/complete the survey. Authors should include this in their limitations section.

Reviewer #2: This is a very important study on the maternity experiences of a vulnerable population of Afghan women. The manuscript is well written and formatted. Although the study was generally well conducted, given that the researchers were working with vulnerable populations, there are a few important details that are missing from the manuscript. My specific comments are as follows:

Methods:

• You used a decolonial feminist approach to visualize participant lived experiences. Given that you interviewed Afghan women, does approach align with their mindset? Was this approach also used to analyse the data from the health professionals and community members?

• Lines 147-148: Please indicate the exact period when the webcomic was developed and tested

• Lines 146-159: Please indicate the criteria that were used to select the 11 Afghan women that were interviewed e.g Afghan women who had recently delivered in Serbia. Please also briefly profile the women that were involved in the production of the comic. It is also not clear if the 5 participants that co-developed the comic were similar or different from the 6 other participants that only participated in the interview

• The study interviewed Serbian CSOs. Do Afghan civil society groups (formal or informal) exist in Serbia? If yes, then it would have also been good to include them in the study.

• Lines 219-228: In the introduction you indicated that registered Afghan refugees can access a range of healthcare services including maternity care. Unregistered refugees living in squatter camps can only access emergency care. Given the different types of healthcare services registered vs unregistered refugees can access, it would be good for you to describe the types of healthcare professionals that were interviewed in this study. Did these healthcare professionals work in maternity wards? Adding this information will contextualize the results

• Line 242: In the methods section you indicated that you used reflexive thematic analysis and that ES developed the themes. Did the other authors also analyse the data and interpret the findings? Were the same researchers involved in the development of the web-comic and analysis of the data?

Discussion

• Line 425-428: This study aimed to understand the awareness and attitudes of Serbian health professionals and members of women’s civil society organisations about the perinatal experiences of Afghan women in Serbia, using a webcomic to elicit responses. Therefore I suggest that you move line 434 to the first few sentences of the discussion (lines 425-428) as it highlights that most of the respondents were not aware of Afghan women's maternity experiences

• lines 513-521: There is potential selection bias because respondents needed access to an online survey. This study only asked questions on the healthcare experiences of Serbian women. Based on the results, it seems some of poor healthcare practices are experienced by all women in Serbia. Therefore the study could have benefited from asking general questions on maternity care in Serbia to contextualize the results

• Limited information is provided on the study participant characteristics. Additional information on who was included in the study (e.g. type of healthcare professional, maternal age, pregnant or postpartum woman etc.). would contextualize the results

6. PLOS authors have the option to publish the peer review history of their article (what does this mean?). If published, this will include your full peer review and any attached files.

**Do you want your identity to be public for this peer review?** For information about this choice, including consent withdrawal, please see our Privacy Policy.

Reviewer #1: **Yes: **Hanani Tabana

Reviewer #2: No

---

## [Editor Report · Decision Letter 1]

17 Jan 2024

“When a story gets a face…”:  visual elicitation of Serbian perspectives on Afghan refugee women’s maternity experiences in Serbia.

PGPH-D-23-00818R1

Dear Dr Howard,

We are pleased to inform you that your manuscript '“When a story gets a face…”:  visual elicitation of Serbian perspectives on Afghan refugee women’s maternity experiences in Serbia.' has been provisionally accepted for publication in PLOS Global Public Health.

Best regards,

Hannah Tappis, DrPH, MPH

Academic Editor